# Biological and Cosmetical Importance of Fermented Raw Materials: An Overview

**DOI:** 10.3390/molecules27154845

**Published:** 2022-07-28

**Authors:** Weronika Majchrzak, Ilona Motyl, Krzysztof Śmigielski

**Affiliations:** 1Department of Environmental Biotechnology, Faculty of Biotechnology and Food Sciences, Interdisciplinary Doctoral School, Lodz University of Technology, 171/173 Wólczańska Street, 90-924 Lodz, Poland; 2Department of Environmental Biotechnology, Faculty of Biotechnology and Food Sciences, Lodz University of Technology, 171/173 Wólczańska Street, 90-924 Lodz, Poland; ilona.motyl@p.lodz.pl (I.M.); krzysztof.smigielski@p.lodz.pl (K.Ś.)

**Keywords:** bioferments, natural cosmetics, fermentation, bioactive compounds

## Abstract

The cosmetics industry is currently looking for innovative ingredients with higher bioactivity and bioavailability for the masses of natural and organic cosmetics. Bioferments are innovative ingredients extracted from natural raw materials by carrying out a fermentation process with appropriate strains of microorganisms. The review was conducted using the SciFinder database with the keywords “fermented plant”, “cosmetics”, and “fermentation”. Mainly bioferments are made from plant-based raw materials. The review covers a wide range of fermented raw materials, from waste materials (whey with beet pulp) to plant oils (F-Shiunko, F-Artemisia, F-Glycyrrhiza). The spectrum of applications for bioferments is broad and includes properties such as skin whitening, antioxidant properties (blackberry, soybean, goji berry), anti-aging (red ginseng, black ginseng, *Citrus unshiu* peel), hydrating, and anti-allergic (aloe vera, skimmed milk). Fermentation increases the biochemical and physiological activity of the substrate by converting high-molecular compounds into low-molecular structures, making fermented raw materials more compatible compared to unfermented raw materials.

## 1. Introduction

Natural cosmetics include a wide range of products for skin care, hair care, color cosmetics, and fragrance products [1]. Organic and natural cosmetics are qualitatively similar to conventional cosmetic preparations, but are chemically and technologically different [2,3]. In order for a cosmetic to obtain natural or organic status it must be certified. There are no standardized guidelines from the organizations that certify products as organic and natural. The most recognized certification governments are Ecocert, COSMOS, and the FDA [4]. During the production of natural and organic cosmetics, it is necessary to refrain from processes such as the use of genetically modified raw materials, radioactive irradiation for decontamination of raw materials, and the final product. It is also forbidden to use synthetic fats, parabens, artificial colors and flavors, silicones, paraffin and other petroleum derivatives and PEG emulsifiers [5,6,7,8].

An alternative approach of the cosmetics industry is to use fermentation processes to obtain biomolecule-rich cosmetic ingredients. The raw materials produced in this way are commonly called “*Bioferments*” [9]. Specific microbial strains used in the fermentation process produce additional beneficial compounds such as proteins, ceramides, amino acids, and antioxidants, thanks to which the products show increased biological effectiveness and bioavailability [10,11,12]. Cosmetics containing bioferments in their formulation are increasingly marketed as natural or more health-friendly cosmetics. Surveys show that users are familiar with these types of products as cosmetics that do not use synthetic chemicals. A survey of a group of people who tested natural cosmetics and cosmetics products with fermented ingredients showed 90% positive results for products with fermented ingredients and 50% for natural cosmetics. The study showed that cosmetics containing fermented ingredients are innovative and can have a competitive advantage on the cosmetics market [13].

Bioferments have been used not only in skin care cosmetics, but also in hair care. An example of this is an anti-itch hair dye which contains a bioferment from plant biomass with the following composition: *Aloe vera*, *Rohdea japonica*, papaya, *Lonicera japonica* flower, tea seed meal, lemon, pomelo, and ginger peel. The innovative hair dye has no toxic effects on the condition of hair and scalp. It reduces greasy hair and moisturizes the scalp, which is helpful in the fight against itching [14].

The review was conducted using the SciFinder database with the keywords “fermented plant”, “cosmetics”, and “fermentation”. The graph (Figure 1) clearly shows that the significant increase in publications with a mention of bioferments in cosmetics was in 2013. This means that the topic is relatively new and the trend related to bioferments only started in 2021.

## 2. Overview of Bioferments

### 2.1. Microbial Products in Cosmetics

The major advantage of cosmetic ingredients derived from fermentation by microorganisms is their biocompatibility. This also has an implication in environmental and technological aspects such as reduction of carbon footprint, use of plant biomass wastes, and new technology for raw material extraction. Fermentation is an effective process of processing raw material to enrich its matrix with biologically active compounds, but the degree of its use in the cosmetics industry is still minor [15].

Microorganisms are the source of compounds such as cyclodextrins, which are used to stabilize the fragrance in a cosmetic product. These compounds are mostly produced by biosynthesis, mainly by *Bacillus subtilis* 313, *Brevibacterium* sp. Strain 9605, *Brevibacillus brevis* CD 162, and *Microbacterium terrae* KNR 9 [16]. An interesting aspect is also the production of biosurfactants by strains such as *Pseudozyma antarctica*, *Pseudozyma aphidis*, *Pseudozyma rugulosa*, and *Pseudozyma parantarctica*, as well as *B. subtilis*, *B. pumilus* A, *B. licheniformis*, and *B. amyloliquefaciens* [17]. Among the largest percentage of biosurfactants are compounds such as fatty acids, glycolipids, neutral lipids, and lipopeptides. Rhamnolipids, which are also biosurfactants, have been approved by the US Environmental Protection Agency (EPA) as safe for use in cosmetic formulations [17,18]. *Candida albicans*, *Agaricus bisporus*, and *Armillaria tabescens* are mainly used to produce glycosylides. Using metabolic engineering techniques, ceramides of microbial origin are obtained by the strain *Saccharomyces cerevisiae*. Florotannin extracts from the seaweed *Eisenia bicyclis* and the *Ecklonia cava* (brown algae) have a very good effect on skin condition by significantly reducing elastase activity. Green microalgae, including *Chlorella*, restore skin firmness while protecting collagen and elastin fibres from the damaging effects of enzymes, collagenase and elastase. Brown algae *Macrocystis pyrifera* extract promotes the synthesis of hyaluronic acid by stimulating the synthesis of syndecan-4 in the extracellular matrix of skin tissues [19].

Fermentation increases the biochemical and physiological activity of the substrate by converting high-molecular compounds into low-molecular structures, making fermented raw materials more compatible compared to unfermented raw materials Figure 2 [20].

The fermentation process depends mainly on the choice of microorganism strain and the conditions under which the process is carried out. The selection of suitable microorganisms is very important in terms of obtaining a suitable matrix of biologically active compounds [21,22,23]. Table 1 describes bioferments and their cosmetic properties.

### 2.2. Soybean

The soybean milk fermented by the *B. breve* strain has a significant increase in hyaluronic acid (HA) production, as demonstrated in a cellular test on skin cells (keratinocytes). Hyaluronic acid, which contributes to skin hydration and elasticity, is naturally produced in human keratinocytes and fibroblasts [24,25]. The *B. breve* Yakult strain has the ability to convert isoflavone glycoside from soya into aglycone via β-glucosidase [26,27]. Fermentation of black soybean and black soy milk with sucrose (2%) or skimmed milk was carried out with *Lactobacillus plantarum* WGK 4, *Streptococcus thermophilus* Dad 11, and *Lactobacillus plantarum* Dad 13 at 37 °C for 18 h. The results showed that the strains could grow (9 log cfu ml^−1^) and produce lactic acid in black soybean and black soy milk with 2% sucrose or skimmed milk. Fermented black soy milk without added sucrose or skimmed milk showed the highest β-glucosidase activity and antioxidant capacity, indicating that all three strains of lactic acid bacteria increase the antioxidant activity of black soy milk. The addition of sucrose or skimmed milk had no effect on cell growth, but increased acid production and decreased β-glucosidase activity [28].

Soybean aqueous extract is also used to produce bioferments by *Bacillus subtilis* strains [29]. The soybean aqueous extract was fermented by *Bacillus subtilis* strain 168. The fermented (FSB) and unfermented extract (SB) were then extracted by reflux-condensation in 80% ethanol [30]. The antioxidant and bleaching activities of both starting materials were evaluated by determining DPPH (spectrophotometrically), superoxide radical and hydroxyl radical scavenging activity, and linoleic acid inhibition and tyrosinase inhibition activity. Optimal fermentation conditions were also determined by measuring the DPPH radical scavenging activity. FSB at 40 °C had significantly higher antioxidant activity than FSB at 30 °C. The highest free radical scavenging effect was 83.5% after 36 h. The DPPH scavenging activity of FSB was 20% higher than that of SB, and similar results were obtained when comparing FSB with vitamin C. FSB showed higher superoxide and hydroxyl radical scavenging activity than SB and vitamin C. FSB showed higher linoleic acid inhibitory activity than SB, but lower than vitamin C [30,31,32]. Therefore, it is possible to attribute the whitening and skin tone levelling properties to FBS, as linoleic acid affects skin pigmentation. The tyrosinase inhibition test showed the best results for FSB, proving that it can play a role in skin bleaching by inhibiting melanin production and confirming that antioxidants are good tyrosinase inhibitors [33,34,35].

### 2.3. Skimmed Milk

The moisturizing effect on the skin is the main purpose of fermenting the ingredients, as lactic acid bacteria and bifidobacteria produce lactic acid and amino acids, which are the main components of natural moisturizing factor (NMF). The supernatant of fermented skimmed milk using *Streptococcus thermophilus* ST-1 (SE), the supernatant of fermented skimmed milk using *Streptococcus*
*thermophilus* YIT 2084 (SE2) [36], and the supernatant of skimmed milk fermented by *Streptococcus thermophilus* YIT 2001 (SE3) are characterized by such properties. Proline-rich peptides have an antioxidant effect and slightly acidify the pH, which is important for keeping the skin in good condition by preserving the residual bacterial flora. The supernatant of skimmed milk fermented by SE2 has a similar effect to the bioferment from SE3. However, it has a greater skin moisturizing effect as it contains hyaluronic acid, which has a strong moisturizing effect along with other moisturizers. It is believed that there is a synergistic effect between hyaluronic acid, lactic acid and amino acids as moisturizing agents in the bioferment [37]. HA is identified by receptors, such as CD44 and the receptor for hyaluronan-mediated motility (RHAMM), and can release various cellular signals, including those associated with wound healing. In addition to triggering cellular signals, HA can attach to cells to form a barrier that prevents oxidative stress [38]. Among lactic acid bacteria cultures (*S. thermophilus* MD2, *L. helveticus* V3, *L. rhamnosus* NS6, *L. rhamnosus* NS4, *L. bulgaricus* NCDC 09, *L. acidophilus* NCDC 15, *L. acidophilus* NCDC 298, and *L. helveticus* NCDC 292), the angiotensin-converting enzyme inhibition profile and the anti-microbial profile of skimmed milk fermented with lactic acid bacteria were also determined. The strains *L. bulgaricus* NCDC 09 and *S. thermophilus* MD2 reduced pH to a very high degree, and acid production was higher in the strains *L. bulgaricus* NCDC 09 and *L. helveticus* V3 than in the other strains. The milk fermented by *S. thermophilus* (MD2) showed the strongest inhibition of angiotensin-converting enzyme. The antimicrobial activity of cell-free supernatant of milk fermented by *S. thermophilus* MD2 and *L. helveticus* V3 was the greatest [39,40].

### 2.4. Ginseng

The fermentation product of black ginseng using *Saccharomyces cerevisiae* (FBG) was investigated as an anti-wrinkle and skin whitening agent. FBG was tested by an in vitro assay using human fibroblasts (HS68). FBG showed no cytotoxicity at concentrations < 10 μg/mL. The bioferment was found to be eye safe at concentrations up to 100 μg/mL. At concentrations between 0.3 and 10 μg/mL, it significantly increased the expression level of type I procollagen in human fibroblasts [41]. A clinical study (23 subjects) was conducted in which the effect was evaluated at day 0, after 4 weeks, and after 8 weeks. The FBG present in the cosmetic formulation showed anti-wrinkle and skin whitening effects, without side effects. The anti-wrinkle effect of FBG results in induction of type I collagen synthesis and inhibition of MMP-1 activity. FBG reduces melanogenesis by inhibiting tyrosinase and reduces oxidation of DOPA, leading to skin whitening properties [42].

The anti-aging potential of red ginseng (RG) and fermented red ginseng was evaluated using *Lactobacillus brevis* (FRG), which were used as ingredients of cosmetic products. Concentrations of polyphenols, flavonoids, uronic acid, and antioxidant activity were significantly higher in bioferment. The ginsenoside content was not significantly different, whereas the content of ginsenoside metabolites was three times higher in FRG (14,914.3 μg/mL). The content of ginsenoside metabolites such as Rg3, Rg5, and Rk1; compounds K, Rh1, F2, and Rg2 [43]; and flavonoid content were increased as a result of fermentation. The activity of tyrosinase inhibitory (IC50) in the bioferment was 27.63 μg/mL and was also stronger than in RG (34.14 μg/mL). The elastase inhibitory activity (IC50) was also higher in the bioferment by as much as 117.07 μg/mL than in RG. In the skin irritation test with 10% RG and 10% FRG, both cosmetic ingredients were found to be virtually non-irritating. In the skin allergen test, the RG group showed a 100% sensitization index and a mean irritation score, while the FRG group showed 20%. In conclusion, FRG offers significantly improved anti-wrinkle and whitening properties and less toxicity compared to RG [44].

### 2.5. Aloe Vera

Aloe vera has antiviral, antimicrobial, wound healing, and anti-inflammatory properties [45]. The supernatant of *Aloe vera* after fermentation by *Lactobacillus plantarum* YIT 0102 shows a huge skin moisturizing effect of up to 400%. *L. plantarum* has the ability to convert malic acid by malolactic fermentation into lactic acid. The combination of fructose and lactic acid significantly increased the water content of the skin. Based on the results obtained from the amino acid analysis, there was a significant increase in serine, alanine, and glycine [46], the main amino acid components of the NMF, which translates into better care properties of the aloe vera bioferment [47]. The use of aloe fermentation significantly accelerates the healing of burn injuries by reducing inflammation and modifying the microflora. In vivo results showed that aloe bioferment positively influences the production of eosinophils and proliferation of fibroblasts. A positive, noticeable effect of the bioferment on burns was observed already after 14 days. ELISA results showed that fermentation of aloe vera considerably reduced the production of pro-inflammatory factors TNF-α and IL-1β and significantly increased the yield of the anti-inflammatory factor IL-4 in animal serum [48]. Tests on human fibroblast cell lines have proven that fermentation of aloe vera gel with the Lactobacillus plantarum strain improves the anti-wrinkle properties of the raw material. Treatment of human fibroblasts with aloe vera bioferment at a dose of 0.5% increased collagen production by 170% and inhibited MMP-1 synthesis by 48% due to its very high antioxidant activity. The bioferment showed a matrix dominated by polysaccharides of relatively low molecular weight (MW): 20% of all polysaccharides in the bioferment ranged from 600 to 900, while 95% of all polysaccharides in the unfermented aloe vera gel were in the MW range of 200,000 to 300,000. This is because the polysaccharide molecules in the extract are broken down during the fermentation process, improving penetration and in turn anti-wrinkle activity [49].

A group of 35 patients with aphthous stomatitis (RAS) were included to investigate the potential advantages of aloe vera bioferment. Results indicated that the wound healing time after aphthae of the group using bioferment showed potentially better healing results at 4–6 days of treatment. Furthermore, the use of aloe vera fermentation gel can restore the natural oral microbiome and reduce the number of harmful oral bacteria, including *Actinomyces*, *Granulicatellai*, and *Peptostreptococcus*. These results show that aloe vera fermentation gel has positive prospects in clinical applications [50].

### 2.6. Berries

Exposure to ultraviolet (UV) radiation causes sunburn, inflammation or photo-aging of human skin, which is associated with reduced collagen synthesis. Blackberry fermented by *Lactobacillus plantarum* (FBB) was tested on non-human frontal skin fibroblasts (Hs68) and on hairless SKH-1 mice. In this study, blackberry bioferment was shown to reduce wrinkle formation on dorsal skin and induce epidermal thickening in hairless mice irradiated with UVB. The effect of bioferment inhibited UVB-induced reduction of type I procollagen and inactivation of an antioxidant enzyme, indicating that FBB exhibits anti-ageing effects by inhibiting UVB-induced degradation of type 1 procollagen, expression of matrix metalloproteinase (MMP)-1 and MMP-2 proteins, and inhibition of nuclear factor κB (NF-κB) activation and phosphorylation of mitogen-activated protein kinase (MAPK) at Hs68. These factors are protein complexes that act as transcription factors and are involved in the cell’s response to stimuli such as stress, free radicals, and ultraviolet radiation [51].

Goji berries have compounds such as polysaccharides, carotenoids, betaine, phenols, and flavonoids [52]. Different bacterial strains SLV (*Lactobacillus rhamnosus, Lactobacillus reuteri and Bacillus velezensis*), SZP (*Lactobacillus rhamnosus, Lactobacillus plantarum and Bacillus licheniformis*), and SZVP (*Lactobacillus rhamnosus, Lactobacillus plantarum, Bacillus velezensis and Bacillus licheniformis*) have been used to ferment goji berries. Total lactic acid concentration was highest in the bioferment that was inoculated with SZVP. Fermentation soluble protein concentrations incremented by 90.54%, 31.18%, and 110.91% in SLV, SZP, and SZVP, respectively, while soluble protein content was higher in bioferments with SZVP and SLV, which may be associated with the attendance of *B. velezensis* and *L. reuteri* [53]. *B. velezensis* produces bacteriostatic substances containing cyclic lipoprotein peptides during fermentation, which translates into antibacterial properties. Fermentation increased the concentration of 2-pentyl-furan, which is a product of fatty acid oxidation, by 8.72, 6.44, and 11.63 s for SLV, SZP, and SZVP, respectively. Thus, the aroma profile of the bioferment was improved, enriching the bouquet with floral and grassy aromas. The concentration of total phenols was 467.13 mg/100 mL in unfermented raw material and increased by 1.27, 1.11, and 1.07 s in samples that were fermented with SLV, SZP, and SZVP strains. Flavonoid content differed between the unfermented raw material and bioferments (SZP and SZVP) and was highest in the SLV bioferment. All bioferments contained five types of phenolic acids: p-hydroxybenzoic acid, ferulic acid, caffeic acid, p-coumaric acid, and chlorogenic acid. In the fermented raw material samples, the concentration of free forms of p-hydroxybenzoic acid and chlorogenic acid increased significantly. Fermentation leads to hydrolysis of chlorogenic acid to yield caffeic acid. The total concentration of the free form of p-coumaric acid was 1.81 s higher (SLV), while the other two samples showed no free form of p-coumaric acid, which could be due to the presence of the *L. plantarum* strain. All samples contained flavonoids such as rutin, quercetin, naringenin, luteolin, and kemferol. Compared to the non-fermented raw material, the concentration of the free form of rutin increased 1.35, 1.29, and 3.10 s in the SLV, SZP, and SZVP samples. respectively. The degree of DPPH free radical scavenging increased with increasing fermentation time [53,54,55,56,57].

### 2.7. Agro-Food Waste

The use of fermentation refers not only to improving the bioavailability of the raw material and refining its matrix of biologically active compounds, but also to producing compounds with aromatic qualities. Twenty-eight yeast strains were isolated from food and plant samples, and the possibility of producing 2-phenylethanol (2-PE) in organic waste-based media supplemented with L-phenylalanine (L-Phe) was evaluated. 2-phenylethanol (2-PE) is an alcohol with a pleasurable rose scent that is widely used in the cosmetics industry as a flavoring compound and/or preservative. To carry out the fermentation process, whey was used as the media base and by-products from sugar beet processing as the carbon source. Among the isolated strains, eight producing 2-PE in the highest range during 72 h batch cultures were selected. The strains were ascribed to the following species: *Meyerozyma caribbica*, *Metschnikowia chrysoperlae*, *Meyerozyma guilliermondii*, *Pichia fermentans*, and *Saccharomyces cerevisiae*. *S. cerevisiae* is a well-known and promising producer of 2-phenylethanol, while the other listed microorganisms are still poorly studied for this application. The conclusion is that yeasts are sources of 2-PE, but microbial synthesis is not yet economically justified in relation to the chemical syntheses that have been developed [58].

Aqueous extracts of *Citrus unshiu* peel were fermented with *Schizophyllum commune* QG143 and the biological activity of the bioferment was investigated. The content of naringenin (polyphenol) and hesperetin (flavone) increased to 319.8 and 134.9 μg/mL after 84 h of fermentation, a more than tenfold increase in the biological activity of these compounds. In contrast, the content of narrutin and hesperidin was markedly reduced after fermentation by *S. commune* under the same conditions. Such results can be attributed to the hydrolysis process and its use as a carbon source for mycelial growth and metabolite production. An ELISA method was used to evaluate the effect of unfermented *C. unshiu* peel (CPE) and fermented *C. unshiu* peel with *S. commune* (S-CPE) on MMP-1 expression in UVA-irradiated human dermal fibroblasts (HDF). Fibroblasts that were irradiated with UVA after treatment with S-CPE decreased MMP-1 expression by 50.4%, 64.6%, and 81.3% at bioferment concentrations of 0.025, 0.05, and 0.1% (*w*/*v*), respectively, while the non-fermented extract showed negligible activity in this regard. Moreover, the inhibitory effect at a bioferment concentration of 0.1% (*w*/*v*) was relatively greater than that of trans retinoic acid (tRA), which is a well-known UVA-induced MMP inhibitor [59].

### 2.8. Herbal and Plant Mixtures

Natural herbal ingredients are commonly used in cosmetics to enhance properties such as moisturizing or whitening the skin. *Dendrobium officinale* contains compounds such as alkaloids, phenanthrene, and polysaccharides. The extract has advantageous dermal coating properties, offering potential for cosmetic use [60]. Using yeast, Lactobacillus, and Bacillus subtilis mixed in a 1:1:1 proportion, fermentation of *Dendrobium officinale* was carried out. The fermentation effects of the mixed strains were better than those of a single strain. There is a synergistic effect between the different strains using the mixed bacteria for fermentation, meaning that the end result of the mixed fermentation is better than the fermentation carried out with an isolated strain. Compared to a single strain, mixed cultures can lead to advantages such as increased productivity, better substrate utilization, higher adaptability to changing conditions, and high resistance to contamination by unwanted microorganisms. Using a mixed culture of bacteria is not only beneficial in terms of more efficient fermentation, but also carries economic benefits. These stem from the lack of need to isolate a single strain, which reduces the cost of the overall process and reduces working time [61]. Total polysaccharides decreased, while total polyphenols and flavonoids increased in the bioferment from *Dendrobium officinale*. With increasing fermentation time, the degree of free radical scavenging (OH, DPPH, ABTS) and tyrosinase inhibition and UVB absorption by the bioferment initially increased (0–3 days), and then showed a decrease on subsequent days. *Dendrobium officinale* bioferment is better than non-fermented raw material in terms of bioactivity, antioxidant activity, tyrosinase inhibition, and UVB absorption. The bioferment has great potential as a innovative cosmetic raw material. The UV spectra of unfermented *Dendrobium officinale* extract and bioferment are similar, with the main absorption bands being in the 280~320 nm and 360~400 nm ranges, meaning that bioferment can absorb both types of radiation (UVB and UVA). As fermentation increased, the SPF value of the bioferment increased, as shown by statistical tests [62].

Extracts from walnut, asparagus root, and Moutan Cortex Radicis (MCR) were fermented with *Bacillus bifidum*. The bioferments differed not only in the solvents used for extraction (95% ethanol, 50% ethanol, 100% acetate, and 50% acetate) but also in the fermentation time. Their cytotoxicity was assessed by analyzing the viability of human skin fibroblast cells, CCD-966SK, and mouse melanoma cells, B16F10. The 50% ethanolic extracts from herbs showed the greatest phenolic content and tyrosinase inhibitory and antioxidant activity. The phenolic activity of the fermented extracts was significantly higher than that of the unfermented extracts. The optimum system tyrosinase inhibition values for fermented extracts of walnut and asparagus were 420, 380 and 260 μg/ml, or more. Even at 300–900 μg/mL, fermented extracts tested were non-cytotoxic to both CCD-966SK and B16F10 cells. All fermented herbal extracts and asparagus root extract (50% ethanol) fermented with *B. bifidum* for 24 h proved to be the skin whitening product with the greatest antioxidant potential [63].

Another bioferment is also extracted from a mixture of herbs fermented by *Phellinus linteus*. The aqueous extract included *Glycyrrhiza glabra*, *Broussonetia kazinoki*, *Angelica gigas*, *Atractylodes macrocephala*, *Poria cocos*, *Morus alba* (root bark), *Paeonia albiflora,* and *Lithospermum officinale*. Bioferment has been shown to inhibit melanogenesis by activating the phosphatidylinositol 3-kinase/Akt/glycogen synthase kinase-3beta signalling pathway, and down-regulates microphthalmia-associated transcription factor. This clearly indicates that bioferment is an excellent raw material for cosmetic formulations that are not only intended to whiten the skin, but also to treat hyperpigmentation [64].

**Table 1 molecules-27-04845-t001:** An overview of fermented raw materials with cosmetic properties.

Raw Material	Microorganisms	Properties	References
Blackberry ^1^	*Lactobacillus plantarum*	Antioxidant, skin whitening	[11]
*Fructus arctii* ^1^	*Grifola frondosa* HB0071	Anti-ageing, anti-wrinkle	[15]
*Astragalus membranaceus var. mongholicus* ^1^	*Bacillus subtilis* natto, *Bacillus subtilis* ATCC 7059	Anti-ageing, anti-inflammatory, and skincare	[15]
Red ginseng ^1^	*Lactobacillus brevis* or *Saccharomyces cerevisiae*	Anti-wrinkle, skincare, anti-inflammatory, and anti-allergenic	[15]
*Prunus persica** (L.) Batsch*, *Paeonia suffruticosa Andr*., and *Asparagus cochinchinensis (Loureiro) Merrill*^2^	*Bifidobacterium bifidum*	Antioxidant, skin whitening, and reduction of discoloration	[15]
Black ginseng ^1^	*Saccharomyces cerevisiae*	Anti-wrinkle, antioxidant	[15]
*Codonopsis lanceolata* ^1^	*Lactobacillus rhamnosus* GG	Skin whitening, reduction of discoloration	[15]
clove, black galingale, betel, noni, green tea, and mangosteen ^3^	*Lactobacillus plantarum*	Anti-inflammatory, promotes wound healing	[15]
Soybean ^1^	*Bifidobacterium animalis* *Saccharomyces cerevisiae* *Bacillus subtilis*	Improve skin hydration and elasticity, antioxidant, skin whitening, and reduction of discoloration	[15,21,30]
*Citrus unshiu* peel ^1^	*Schizophyllum commune* QG143	Anti-ageing, anti-photo-aging	[15,59]
Wasabi root ^1^	*Lactobacillus*	Antimicrobial, preservation, and antioxidant, anti-inflammatory	[21]
Lemon peel ^1^	*Lactobacillus lactis*	Skin whitening, reduction of discoloration, and antioxidant	[21]
Tonka bean ^1^	*Lactobacillus*	Anti-ageing, anti-photo-aging	[21]
Mannitol and maltodextrin:	*Lactobacillus*	Protects against allergens, free radicals, photo-aging, and pollution; increases skin regeneration; strengthens the protective barrier of sensitive skin; reduces redness and irritation; bacteriostatic effect against skin pathogens such as *Staphylococcus aureus*	[21]
Carrot root ^1^	*Bacillus ginsengisoli*	Antioxidant, improving skin dullness, anti-ageing, and anti-wrinkle	[21]
Radish root ^1^	*Lactobacillus casei*	Antioxidant, heals damaged cells, clears toxins, nourishes the skin, and anti-ageing	[21]
Sea kelp ^1^	*Lactobacillus*	Natural film, excellent oil-free moisturizer	[21]
Skim milk	*S. thermophilus* YIT 2001, *S. thermophilus* YIT 2084	Skin hydration, antioxidant, and maintains proper pH of the skin	[26]
Aloe vera ^1^	*L. plantarum* YIT 0102	The greatest (400%) skin hydration effect	[26]
Soybean milk	*Bifidobacterium breve* Yakult	Improves skin hydration and elasticity	[26]
Goji berry (*Lycium barbarum* L.) ^1^	*Lactobacillus rhamnosus, Lactobacillus reuteri Bacillus velezensis* or *Lactobacillus rhamnosus, Lactobacillus plantarum, Bacillus velezensis Bacillus licheniformis*	Antioxidant, skin whitening, and anti-ageing	[53]
Walnut, Moutan Cortex Radicis and asparagus root ^4^	*B. bifidum*	Antioxidant, skin whitening, and anti-photo-aging	[62]
*Glycyrrhiza glabra*, *Broussonetia kazinoki*, *Angelica gigas*, *Atractylodes macrocephala*, *Poria cocos*, *Morus alba* (root), *Paeonia albiflora* and *Lithospermum officinale* ^1^	*Phellinus linteus*	Skin whitening, reduction of discoloration	[63]
*Camellia sinensis* (black, green, and white tea) ^1^	*Alcaligenes piechaudii* CC-ESB2	Antioxidant, skin whitening, and reduction of discoloration	[64]

^1^. Solvent: water. ^2^. Solvent: 50% ethanol, 95% ethanol, 50% ethyl acetate and water. ^3^. Solvent: 0.1–3.0% of peppermint oil. ^4^. Solvent: 50% ethanol.

### 2.9. Cosmetic Kombucha

Kombucha is a drink made by fermenting tea with sugar using a symbiotic culture of bacteria including to genus *Acetobacter*, *Gluconobacter*, and yeast of the genus *Saccharomyces* together with glucuronic acid [65,66,67]. Bioferments (*Yerba Mate* with *Kombucha)* after different fermentation times are presented. The extracts that were fermented and those that were not fermented were compared in terms of chemical composition and biological activity. Cytotoxicity was determined by cellular assays on keratinocyte and fibroblast lines, which showed a significant increase with bioferments. Bioactive compounds including phenolic acids, xanthines, and flavonoids were identified in the composition of bioferments. The predominant compounds were the isomers caffeoylquinic acid (CQA) and dicaffeoylquinic acid (diCQA). The samples tested also contained a considerable amount of flavonoids, rutin, and caffeine. The total content of phenolic compounds in the studied samples was highest for the bioferment after 21 days of fermentation compared to the unfermented *Yerba Mate* extract (YM). Increasing the fermentation process time to 35 days resulted in a significant reduction in polyphenol content. The bioferments had higher antioxidant activity than the unfermented *Yerba Mate* extract, results from the production of post-fermentation low molecular weight components, and the modification of tea polyphenols by enzymes produced by bacteria and yeast during fermentation. The extracts analyzed showed various antioxidant properties. It was observed that after a time of 10 min, the YM extract slowed down its purification capacity compared to the rest of the analyzed samples. ABTS radical neutralization analysis showed that the most favorable values were observed for ferments obtained after 21 days. The value obtained in this case was the lowest by about 2% compared to *Yerba Mate* extract. The presence of bacteria and yeast released during the fermentation process results in a lower IC50 for Kombucha-fermented extracts, which may result in better efficiency against nitrogen and superoxide radicals but poorer purification efficiency against hydroxyl radicals. Analysis of the reduction of intracellular production of reactive oxygen species in two skin cell lines, fibroblasts (BJ) and keratinocytes (HaCaT), showed that both YM extract and the obtained ferments could reduce intracellular free radical levels in both fibroblasts and keratinocytes. Significant reductions in free radical levels for fibroblasts and keratinocytes were demonstrated for the ferments after 14 and 21 days of fermentation. Neutral Red test results showed the same correlations as the previous analysis. Additional analyses showed that *Yerba Mate* extract at the highest concentration (1000 μg/mL) induced a cytotoxic effect on fibroblasts. However, in the case of HaCaT cells, this effect was observed for the two highest concentrations tested (500 and 1000 μg/mL), indicating that the resulting bioferments have a more beneficial effect on cells than the non-fermented extract. It was also observed that the application of bioferments and *Yerba* extract to the skin improved skin hydration levels, the increase of which depended on the fermentation time. Compared to the control sample, the strongest ability to improve hydration was observed for ferments after 14 and 21 days [67,68].

### 2.10. Bio-Oils

In addition to extracts and plant biomass, natural fermented oils can be obtained by fermentation. Plant oils are excellent emollients rich in biologically active compounds [68]. The following bioferments are available on the market: F-Shiunko (INCI: *Pseudozyma epicola/apricot kernel oil/olive fruit oil/sunflower seed oil/sweet almond oil/(Angelica gigas/Lithospermum erythrorhizon) root extract*), F-Artemisia (INCI: *Pseudozyma epicola/apricot kernel oil/olive fruit oil/sweet almond oil/sunflower seed oil/Artemisia princeps extract ferment extract filtrate*), and F-Glycyrrhiza (INCI: *Pseudozyma epicola/apricot kernel oil/olive fruit oil/sweet almond oil/sunflower seed oil/licorice root extract ferment extract filtrate*). They were introduced as active ingredients in selected cosmetic formulations, and these products were then tested by a group of 20 women. Their cheek skin microbiota was analyzed by swabbing at T0 (before application) and T1 (after 4 weeks). First, the diversity of species richness was analyzed at T0 and T4 on the analyzed skin site. After time T4 of application of cosmetics with fermented oils, a significant increase in diversity was found in all groups based on observations and estimation of OTU abundance and Shannon index values. Analysis of observed OTU abundances, in particular, showed a significant increase in species richness after four weeks of treatment with respect to the zero sample. The treatment also resulted in a more rich and rejuvenated skin microbiome, as treatment with bioferments showed a significant decrease in *Proteobacteria* and an increase in *Staphylococcus*. As a decrease in *Propionibacterium* numbers was observed in the treatment groups, it can be presumed that the fermented oils have a sebum-regulating effect [69,70,71].

## 3. Conclusions

The influence of the fermentation process of plant raw materials on their matrix of biologically active compounds and cosmetic properties is presented. Bioferments in comparison with extracts have increased bioavailability and activity of biologically active compounds. During the fermentation process, complex structures of compounds are converted to simpler forms, and this translates into increased efficiency and bioavailability due to epidermal compatibility and penetration into the skin.

Bioferments help improve the natural balance of the skin’s microbiome, which is disturbed by external factors, which in turn is linked to healthy skin.

Considering the examples of bioferments given in the review, they can be divided into three main groups according to their properties. Bioferments from berries (goi berry, blackberry), soya, lemon peel, and mixtures of herbs are characterized by their antioxidant properties. It has also been observed that antioxidant properties are also strongly associated with whitening and skin tone levelling properties. Another category is anti-ageing and anti-wrinkle properties, which are characterized by bioferments from red and black ginseng, *Cistrus uishu* peel, tonka bean, mannitol, and maltodextrin. The last group consists of bioferments with strong moisturizing properties such as aloe, sea kelp, skim milk, and soybean milk. These raw materials already have excellent skin moisturizing properties before fermentation, while the fermentation process increases these properties by up to 400% (aloe).

Lactic and prebiotic bacterial strains are mainly used to construct bioferments. Yeast strains *Saccharomyces cerevisiae*, which are mainly associated with alcoholic fermentation, are also frequently used, but have been shown to be an excellent microorganism for the construction of bioferments with aromatic qualities.

There is still a lack of research conducted on mixed fermentation, which is carried out by bacteria and yeast. This type of fermentation could yield interesting results given the synergistic action of the microorganisms.

The amount of research carried out on bioferments is quite extensive, while publications on this subject cover areas of the same raw materials or plant extracts. The practical use of bioferments in cosmetic compounds is still negligible; therefore, further technological research in this direction is necessary.

## Figures and Tables

**Figure 1 molecules-27-04845-f001:**
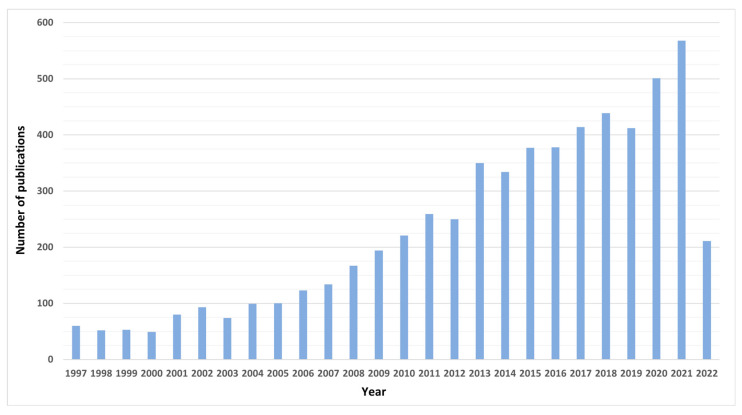
Number of publications in SciFinder from 1997 to first half of 2022.

**Figure 2 molecules-27-04845-f002:**
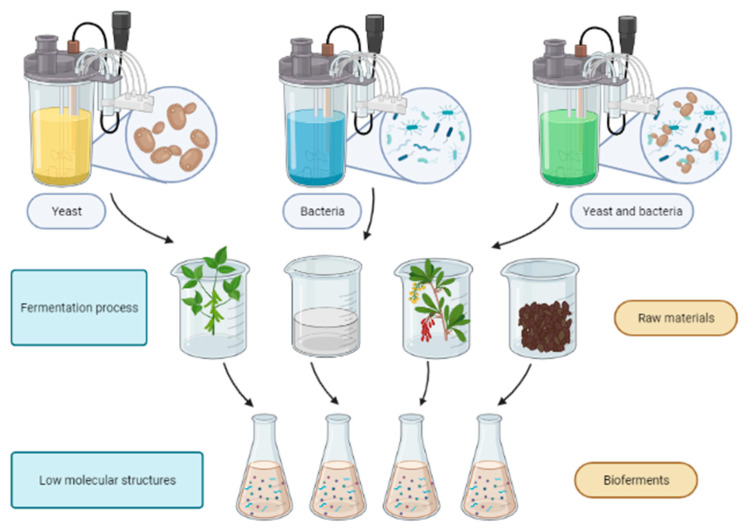
Schematic process of the biofermentation.

## Data Availability

Not applicable.

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
