# Peer review of "Biological and Cosmetical Importance of Fermented Raw Materials: An Overview"

_molecules, 2022, doi:10.3390/molecules27154845_

Round 1

Reviewer 1 Report

This manuscript is discussing the importance of bioferments in cosmetics industry. Bioferments from different plants were presented in this review. I find this topic interesting and presented in a clear language.

1- However, the manuscript is only reporting what other studies have done. It lacks a summary at the end of each section, and a critical review of what the authors think is best method/material to be adopted (and why) based on the reviewed studies.

2- Abstract should include a summary of the article's main findings, and a conclusive statement indicating the main conclusions or interpretations.

3- The novelty of the study needs to be highlighted at the end of the introduction section.

4- in page 5 line 130, "..has are characterized.." please check this sentence.

5- page 8, line 246, "to yield coffee acid" is it caffeic acid?

6- page 8, line 292 "The fermentation effects by the mixed strains were better than those of a single strain.". any reason behind that? also, it is economically favorable since it saves the costs of isolation and purification of the single bacteria.

7- page 9, line 320, please check "..by activating the of the phosphatidylinositol.."

Author Response

Good morning,
The manuscript has been revised with regard to the comments made. 

Yours sincerely,
Weronika Majchrzak 

Reviewer 2 Report

The paper is well written and presents concise yet relevant information for the field. The suggestion I give is to provide more figures to illustrate better the paper, but for the text itself, I have no amendment to suggest.

Author Response

Good morning,
The manuscript has been expanded to include an introduction and a conclusion. 

Yours sincerely,
Weronika Majchrzak  

Round 2

Reviewer 1 Report

The authors have done good job addressing all the comments given.

One last thing, for figure 1 caption, it is better to either not include 2022 or to mention that the number of publications are for the first half of 2022.